# Complex Relationships Between Homologous Recombination Deficiency (HRD) Score and Mutational Status of Homologous Recombination Repair (HRR) Genes in Prostate Carcinomas

**DOI:** 10.3390/ijms262411851

**Published:** 2025-12-08

**Authors:** Aglaya G. Iyevleva, Svetlana N. Aleksakhina, Anna P. Sokolenko, Ekaterina A. Otradnova, Alisa S. Nikitina, Kira A. Kashko, Maria V. Syomina, Anna D. Shestakova, Ekaterina S. Kuligina, Natalia S. Morozova, Sergey V. Popov, Pavel V. Vyazovcev, Tatyana Y. Luchkova, Aleksey S. Peremyshlenko, Timur M. Topuzov, Olga M. Gudkova, Rashida V. Orlova, Andrey V. Levushkin, Daniil O. Moiseev, Oksana N. Shkodenko, Demyan V. Gubarev, Aleksandr V. Dzyuba, Irina Y. Povorina, Anna V. Agaeva, Vladislav F. Kutyan, Mhitar M. Grigoryan, Alexey N. Orlov, Spartak I. Lefterov, Aleksey V. Belousov, Marina N. Nechaeva, Elena N. Vorobyeva, Irina K. Amirkhanova, Nadezhda V. Kryukova, Lyubov I. Zatulivetrova, Aryuna B. Gomboeva, Vladimir N. Krivtsov, Olga I. Shchiglik, Natalya V. Prokudina, Natalya A. Butunina, Aleksey M. Belyaev, Evgeny N. Imyanitov

**Affiliations:** 1N.N. Petrov Institute of Oncology, Saint-Petersburg 197758, Russia; aglayai@inbox.ru (A.G.I.); kate.kuligina@gmail.com (E.S.K.);; 2Department of Medical Genetics, Saint Petersburg State Pediatric Medical University, Saint Petersburg 194100, Russia; 3Saint Luke’s Clinical Hospital, Saint Petersburg 194044, Russialuchkova@lucaclinic.ru (T.Y.L.);; 4Saint Petersburg City Oncological Clinic, Saint-Petersburg 197022, Russia; 5Stavropol Regional Oncological Clinic, Stavropol 355047, Russia; 6Tula Regional Oncological Clinic, Tula 300039, Russia; 7Arkhangelsk Regional Oncological Clinic, Arkhangelsk 163045, Russia; 8Krasnodar Regional Oncological Clinic No. 1, Krasnodar 350040, Russia; 9WMT Clinic, Krasnodar 350063, Russia; 10Leningrad Regional Clinical Hospital, Saint Petersburg 194291, Russia; 11Penza Regional Oncological Clinic, Penza 440071, Russia; 12Severodvinsk City Clinical Hospital No. 2, Severodvinsk 164500, Russia; 13Chelyabinsk Regional Clinical Center of Oncology and Nuclear Medicine, Chelyabinsk 454087, Russia; 14Vologda Regional Oncological Clinic, Vologda 160012, Russia; 15Pyatigorsk Regional Oncological Clinic, Pyatigorsk 357502, Russia; 16Buryat Republican Oncological Clinic, Ulan-Ude 670047, Russia; 17Pskov Regional Oncological Clinic, Pskov 180004, Russia; 18Karelian Republican Oncological Clinic, Petrozavodsk 185002, Russia; 19Kaliningrad Regional Oncological Clinic, Kaliningrad 236016, Russia; 20Saratov Regional Oncological Clinic, Saratov 410053, Russia

**Keywords:** prostate cancer, homologous recombination deficiency, HRD score, mutations in homologous recombination repair genes

## Abstract

Homologous recombination deficiency (HRD) resulting from inactivation of *BRCA1/2* genes promotes chromosomal instability and renders tumor cells susceptible to platinum derivatives and PARP inhibitors (PARPi). The contribution of alterations in other homologous recombination repair (HRR) genes to HRD remains understudied. This investigation aimed to analyze the spectrum of mutations in 34 HRR genes in prostate carcinomas (PCs) and study the relationship between HRR status and HRD. HRR mutations and HRD scores were examined by NGS in 1131 and 680 PCs, respectively. Pathogenic or likely pathogenic variants in HRR genes were detected in 216/1131 cases (19.1%). HRD, defined by an HRD score cut-off of ≥42, was observed more frequently in HRR-mutated than in wild-type tumors (23/120 (19.2%) vs. 29/560 (5.2%), *p* < 0.0001). The highest HRD scores were detected in PCs with biallelic inactivation of the *BRCA2* or *PALB2* genes, as well as in tumors with *BRIP1* mutations. HRD was also occasionally seen in PCs with *ATM*, *NBN, FANCM*, *BRCA1* and *CDK12* alterations, but never in cases with *CHEK2* mutations. HRD was significantly more associated with aggressive PC features than HRR mutations. The majority of *CDK12*-mutated tumors exhibited a distinct type of copy number variations (CNV)–a tandem duplication phenotype. Our study suggests that the selection of PC patients for PARPi treatment requires a significant revision of existing attitudes towards tumor genetic profiling.

## 1. Introduction

Mutations in homologous recombination repair (HRR) genes affect a substantial portion of prostate cancer (PC) cases. The frequency of HRR alterations reaches 20–25% in metastatic castration resistant PC (mCRPC), which is the most aggressive variant of this disease [1,2]. Identifying genetic HRR alterations is becoming increasingly important in clinical practice. Firstly, germline alterations are associated with hereditary predisposition to cancer, providing grounds for genetic testing of the patient’s relatives. Secondly, genetic defects in the HRR system can lead to homologous recombination repair deficiency (HRD), which makes tumor cells susceptible to DNA-damaging chemotherapeutic drugs and synthetic lethality-inducing agents, such as PARP inhibitors (PARPi). For the time being, all four available PARPi (olaparib, rucaparib, niraparib and talazoparib) have been approved for the use in PC patients, although the nuances of their indications vary between drugs [3].

The selection of patients for the treatment with single-agent PARPi relies on the identification of deleterious hereditary or somatic HRR mutations. For instance, single-agent olaparib may be prescribed in cases with a mutation in any of the 15 HRR genes included in the initial clinical trial (*ATM*, *BRCA1*, *BRCA2*, *BARD1*, *BRIP1*, *CDK12*, *CHEK1*, *CHEK2*, *FANCA*, *PALB2*, *RAD51*, *RAD51B*, *RAD51C*, *RAD51D*, *RAD54L*) [4]. The administration of talazoparib relies on another modification of the HRR test, which includes 12 genes (*ATM*, *ATR*, *BRCA1*, *BRCA2*, *CDK12*, *CHEK2*, *FANCA*, *MLH1*, *MRE11A*, *NBN*, *PALB2*, *RAD51C*) [5,6]. Noteworthy, only 8 genes are shared between these 2 panels (*ATM*, *BRCA1*, *BRCA2*, *CDK12*, *CHEK2*, *FANCA*, *PALB2*, *RAD51C*). Furthermore, an increasing amount of clinical and experimental data suggests that distinct HRR genes contribute differently to HRD formation and treatment efficacy. For example, the olaparib study observed a clear positive effect in patients with *BRCA2* mutations, while the presence of *ATM* or some other genetic defects was associated with minimal or no benefit [4,7]. Cell line experiments demonstrated that the loss of *BRCA1/2*, *RAD51*, *XRCC2* or *PALB2*, but not *ATM* or *CHEK2*, results in HRD and, consequently, sensitivity to DNA double-strand breaks inducers (platinum drugs, PARP inhibitors, anthracyclines, and topoisomerase I and II inhibitors) [8]. A recent pooled analysis concluded that PARP inhibitors are beneficial for patients with *BRCA1/2, PALB2* and *CDK12* mutations, but ineffective for PC with *ATM* or *CHEK2* genetic defects [9]. Tumor responses to PARPi or platinum drugs have been reported in patients with *FANCA*, *BRIP1*, *RAD51B* and *RAD54L* mutations; however, this experience remains limited to occasional observations and has not yet been confirmed by the analysis of relevant patient series [4,10,11,12]. According to NGS-based and functional studies, HRD occurs in breast and prostate cancers with mutations in *RAD51C* [13,14], *BARD1* [15] and *RAD51D* genes [16]. However, the available data are scarce and the impact of many other genes involved in HRR processes (*NBN*, *FANCM*, *FANCI*, *FANCC*, *BLM*, *MRE11*, *ATR*, *RAD50*, etc.) on HRD formation or treatment efficacy is still poorly understood.

Genomic HRD signatures appear to be more robust predictors of tumor sensitivity to DNA-damaging therapy when compared to mutations in individual HRR genes. Comprehensive genomic studies of *BRCA1/2*-associated malignancies have revealed chromosomal profiles and patterns of small mutations, which are highly specific for HRD [13,17,18]. Whole-genome sequencing is probably the most reliable approach for identifying the consequences of homologous recombination deficiency; however, it is not yet compatible with routine clinical practice. An acceptable alternative is the analysis of chromosomal aberration profiles with the targeted next-generation sequencing (NGS) SNP panels. In 2016, an HRD score that combines three measures of chromosomal instability, was introduced. It represents a sum of the numbers of genomic LOH regions larger than 15 Mb, large-scale state transitions (LST) (transitions between chromosomal fragments with different copy numbers longer than 10 Mb), and telomeric allelic imbalance (TAI) regions [19]. This score has been shown to be a reliable predictor of tumor response to PARPi and platinum-based chemotherapy in ovarian and breast cancer [19,20,21,22,23]. The diagnostic and predictive role of the HRD score in PC is much less studied [24,25,26]. Some data suggest that *BRCA1/2*-mutated prostate cancers generally have a lower level of chromosomal instability than *BRCA*-associated breast or ovarian cancers [24,27].

The aim of this study was to characterize the spectrum of HRR mutations in Russian prostate cancer patients and to investigate homologous recombination deficiency by determining the HRD score. Additionally, we aimed to compare HRD scores in patients with mutations in various HRR genes.

## 2. Results

### 2.1. Frequency and Spectrum of Mutations in HRR Genes

The coding sequences of 34 HRR genes were analyzed in 1131 prostate cancer cases using targeted NGS analysis. The analysis was performed on paired tumor and normal DNA samples in 947 cases. In 51 patients, sequencing data were obtained exclusively from tumor DNA; in another 133 cases, only blood-derived normal DNA was examined. The clinical and morphological characteristics of the samples are presented in Table 1. Either primary metastatic PC or disease progression after initially localized tumor were diagnosed in 525 patients (46.4% of the total sample or 58.9% of the cases with available clinical information).

Pathogenic or likely pathogenic germline or somatic variants in HRR genes were detected in 216 cases (19.1%), 42 of which had two or three mutations simultaneously. One patient was found to carry a pathogenic germline *TP53* alteration, which is an indicator of Li-Fraumeni syndrome. A total of 262 HRR mutations (excluding the case with *TP53* variant) were identified in 216 patients, comprising 150 germline and 105 somatic alterations. At least one germline alteration was observed in 142 out of 216 patients (65.8%). Sixty-seven cases (31.0%) presented with only somatic mutations. The origin of the mutations, which were identified in tumor tissues, could not be clarified in seven cases (3.2%) due to the absence of corresponding normal DNA.

Mutations were most frequently detected in *BRCA2* (42 out of 216 patients with HRR alterations, 19.4%), *ATM* (42/216, 19.4%), *CDK12* (30/216, 13.9%), *CHEK2* (26/216, 12.0%), *NBN* (15/216, 6.9%), and *FANCM* (12/216, 5.6%) genes. Other common alterations were observed for *FANCC* (9/216, 4.2%), *BRCA1* (8/216, 3.7%), *PALB2* (6/216, 2.8%), *BLM* (6/216, 2.8%), *RAD54L* (5/216, 2.3%), *FANCI* (5/216, 2.3%), *BRIP1* (4/216, 1.9%), *FANCA* (4/216, 1.9%), *BARD1* (3/216, 1.4%), *MRE11* (2/216, 0.9%), and RAD50 (2/216, 0.9%) genes (Figure 1, Appendix A, Appendix A). The majority of *BRCA2* defects were of germline origin (29/41, 70.7%), and three patients harbored concurrent germline and somatic *BRCA2* alterations. Overall, either intratumoral loss of the wild-type allele (LOH) in patients with germline *BRCA2* defects or double *BRCA2* mutations, which presumably result in the complete inactivation of the *BRCA2* gene, were observed in 16 out of 23 (69.6%) informative cases. Germline *ATM* mutations were identified in 17 out of 40 (42.5%) *ATM*-altered cases, while LOH or a combination of somatic and germline *ATM* variants occurred in 16 out of 18 (88.9%) informative cases. The third most frequently affected HRR gene was *CDK12*, which is known to harbor exclusively somatic alterations in PCs. Approximately a half of PCs with *CDK12* variants (16/30, 53.3%) bore double inactivating somatic mutations. The vast majority of mutations in the other HRR genes were of germline origin; LOH of the remaining allele of the involved gene was observed infrequently in these tumors, except for *PALB2* (Appendix A).

The spectrum of *BRCA2* and *ATM* germline variants was highly heterogeneous and contained only few recurrent mutations. In contrast, almost all *CHEK2*, *NBN* and *BRCA1* defects were represented by founder or recurrent variants: 22/26 (84.6%) *CHEK2* alterations belonged to one of the three founder alleles (*CHEK2* c.1100delC, c.444+1G>A and del5395); Slavic founder deletion *NBN* c.657del5 constituted 12/15 (80%) of *NBN* mutations; 6 (66.7%) out of 9 *BRCA1* mutations were recurrent c.3700_3704delGTAAA [3819del5], c.4034delA [4153delA], c.5266dup [5382insC], or c.181T>G [p.C61G] variants. Other founder pathogenic alleles were represented by *FANCM* c.1972C>T (p.Arg658Ter) (n = 4), *FANCI* c.3853C>T (p.Arg1285Ter) (n = 4), *BLM* c.1642C>T (p.Gln548Ter) (n = 4), *FANCC* c.996+1G>T (n = 2), and *BARD1* c.1690C>T (p.Gln564Ter) (n = 2) mutations (Appendix A).

### 2.2. Analysis of HRD Scores

Out of the total 1131 studied cases, HRD testing was successfully completed for 680 patients, for whom the paired tumor and normal DNA of sufficient quality and quantity were available. In these cases, HRD scores were calculated as a sum of LOH, LST and TAI measures obtained after targeted NGS analysis with the HiSNP Ultra Panel v1.0 (Nanodigmbio, Nanjing, China). An HRD score of 42, which is commonly accepted as a clinically valid threshold for ovarian and breast cancer, was used to determine the presence of HRD [19,20,21,22,23].

HRD score equal or higher than 42 was observed in 23/120 (19.2%) PCs with altered HRR genes and in 29/560 (5.2%) tumors from patients without any identified HRR mutations (*p* < 0.0001). The proportion of cases with HRD score ≥ 42 was the highest in tumors with *BRCA2*, *PALB2* and *BRIP1* mutations (50%, 50%, and 67%, respectively, Figure 2B, Appendix A). While only half of *BRCA2*-associated PC had the score above the chosen threshold, the majority of them (14/20, 70%) demonstrated the score very close (≥38) to the predefined point cut-off (≥42). Interestingly, two cases with germline *BRCA2* variants not having *BRCA2* LOH in the tumor showed no signs of chromosomal instability (score = 0). Among 4 analyzed *PALB2*-associated PCs, all three cases with germline alterations accompanied by somatic LOH of the wild-type allele had HRD score higher or very close to the chosen threshold (40, 48, 60), and the case with germline mutation without LOH had chromosomally stable tumor (score = 0) (Figure 2D). High HRD scores were occasionally observed in PCs with *ATM*, *NBN*, *FANCM*, *BRCA1* and *CDK12* mutations, but not in cases with *CHEK2* mutations. Of notice, only 3/12 (25%) PCs with putative *ATM* biallelic inactivation showed high HRD scores (Appendix A).

When analyzed as continuous variable, HRD score in cancers with *BRCA2* and *ATM* alterations was significantly higher than in those without identified HRR mutations. *CHEK2*-associated tumors showed the lowest HRD values and differed in this respect from *ATM-*, *BRCA2-*, *BRIP1-*, or *CDK12*-mutation positive cases (Figure 2C, Appendix A).

The majority of tumors with *CDK12* somatic alterations (11/13, 84.6%) had clearly distinguishable CNV profiles with genome-wide narrow spikes suggestive of a tandem duplicator phenotype [28,29] (Figure 2A). Interestingly, one of the samples categorized as HRR WT showed a *CDK12*-specific CNV profile. Re-analysis of NGS data in the genome browser led to identification of a gross *CDK12* deletion, which was further confirmed by PCR (Appendix A).

### 2.3. Clinicopathological Associations

Neither mutations in HRR genes nor HRD were associated with the age at PC diagnosis. Both HRR alterations and high HRD scores were enriched in PCs with aggressive clinical features, such as high tumor stage and grade (Table 2, Figure 3). These associations were more pronounced for HRD than for HRR status. For example, HRD score ≥ 42 was almost never observed in stage I PC (0.7%) compared to stage IV tumors (16%, *p* < 0.0001). High HRD was also strongly associated with the presence of somatic *TP53* alterations (Table 2, Figure 3).

## 3. Discussion

This study characterized the spectrum of mutations in an expanded list of HRR genes in prostate cancer. The incidence of pathogenic/likely pathogenic HRR alterations in our cohort (19.1%) is slightly lower but close to the numbers previously reported for mCRPC (≈23–28%) [1,2,30,31]. However, when we consider PCs with distant metastases, advanced tumor stage, or high Gleason grade (Table 2), our dataset produced essentially the same results as the mentioned above mCRPC studies. Hereditary variants were identified in the majority (142/216; 66%) of cases with HRR alterations. Of these, 103 patients (8.6% of the total sample) carried germline pathogenic variants in genes that have been definitively linked to an increased risk of PC or other types of cancers (*BRCA2*, *ATM*, *CHEK2*, *NBN*, *BRCA1*, *PALB2*, *BRIP1* and *BARD1*) [32,33,34]. The share of germline variants in the above-mentioned genes reached 9.6% in PC with high Gleason grade and 10.9% in stage III-IV tumors. Approximately one third of the identified germline variants were found in genes with an unclear or yet unstudied relationship to PC predisposition, including *FANCA*, *FANCC*, *FANCI*, *FANCM*, *RAD54L*, *MRE11*, *RAD50*, etc. [35,36]. It is important to acknowledge that the actual frequency of somatic HRR alterations may be somewhat higher than we observed. This is attributable to the fact that only normal DNA was subjected to sequencing in 133/1131 (11.8%) patients included in this investigation.

Analysis of our data in comparison to other studies suggests that there may be interethnic variations in the distribution of HRR mutations in PC. We observed lower incidences of *BRCA2* and *CDK12* mutations (42/1131; 3.7% and 30/1131; 2.7%, respectively) than those reported in large European and American studies, which produced the rates of 9–12% for *BRCA2* and around 7% for *CDK12* [31,37,38]. Conversely, our dataset revealed a significant prevalence of *CHEK2* and *NBN* alterations, accounting for 2.3% and 1.3% of analyzed PCs, respectively. The latter is likely to be a consequence of the high frequency of several founder *CHEK2* and *NBN* cancer-predisposing alleles in Slavic populations. Within our study group, 66% of individuals with HRR mutations had germline alterations. In contrast, Olmos et al. (2025) reported a considerably lower percentage of germline variants (32%) [37]. Along with the distinct ethnic composition of the studied cohorts, these differences could be attributed to the varying clinical characteristics of the patients, non-identical HRR gene panels, testing of either tumor alone or paired tumor and normal tissues, and differences in the interpretation of variant pathogenicity. The reliability of somatic mutation detection also depends heavily on the proportion of tumor cells in the sample and the sequencing depth. We employed a fairly lenient threshold of 10% tumor cells in a tissue sample, which could potentially impact the identification of somatic variants.

An important objective of this study was to compare intratumoral HR status (as determined by HRD score) with the mutational status of various HRR genes, in order to ascertain the association of these genes with HRD-type chromosomal instability. Our results suggest that only a minority (19%) of HRR-mutated PCs have a pronounced homologous recombination deficiency (HRD score ≥ 42), and that the inactivation of different genes is associated with varying degrees of chromosomal instability. Notably, the highest HRD scores were observed in PCs with *BRCA2* (median 41.5), *PALB2* (44) and *BRIP1* (51.0) alterations. An HRD score of at least 42 was found in only a half of *BRCA2*-positive carcinomas. This threshold is clinically significant in ovarian and breast cancer, allowing the detection of 95% of *BRCA1/2*-positive tumors of these types and identifying patients who may benefit from PARP inhibitors and platinum drugs [19,20,21,22,23]. Our data, along with the results of several similar studies, suggest that HRD scores tend to be lower in PCs, including *BRCA2*-associated tumors [24,26,27]. One of the possible explanations is the lower occurrence of somatic *TP53* mutations in PC. Indeed, *TP53* alterations are known to be strongly associated with elevated chromosomal instability [24,26,39], and our study confirmed a strong correlation between *TP53* gene status and high HRD (Table 2; Figure 3). We also examined whether the low HRD scores obtained in *BRCA2*- and *BRCA1*-mutated cases in our study are attributed to suboptimal tumor cell content in the corresponding tissue samples. However, we concluded that this factor was unlikely to have influenced the results in our series, since the *BRCA2*-mutated PC with the lowest observed tumor cell content (15%) demonstrated a high HRD score of 76. Overall, in our dataset 95% of cases with confirmed biallelic inactivation of *BRCA2*, *BRCA1* or *PALB2* had an HRD score of at least 25, whereas only 60% of such tumors demonstrated a score ≥42. Taken together, these data indicate that the optimal threshold for HRD score in PC may indeed differ from the one accepted for ovarian/breast carcinomas. Future studies have to address a predictive significance of various HRD cut-offs for tumor sensitivity to PARPi and other drugs.

Interestingly, we observed two cases with germline *BRCA2* variants and retention of the wild-type *BRCA2* allele in tumors with no signs of chromosomal instability (score = 0), suggesting that the development of these malignancies was not related to *BRCA2*-associated syndrome.

Previous studies have shown that *PALB2* inactivation results in genomic features of HRD [13,14,15,40,41,42]. The results of the PARPi trials also favor the efficacy of PARPi in PC with *PALB2* mutations [4,10,43]. Our data are consistent with these observations, as all three tumors with biallelic *PALB2* lesions had HRD scores above or very close to the threshold. Similarly to *BRCA2*, a case with germline *PALB2* mutation and the absence of somatic LOH had a chromosomally stable tumor (Figure 2D). It can be concluded that for hereditary *BRCA2* and *PALB2* mutations, the intratumoral loss of the wild-type allele is the main mechanism of somatic second hit, resulting in elevated chromosomal instability.

*BRIP1* mutations increase the risk of ovarian cancer, and it has been demonstrated that *BRIP1*-deficient cells are sensitive to combined treatment with platinum drugs and PARPi [44]. The number of observations was small; however, we identified HR deficiency in two out of three cases with *BRIP1* alterations.

This study included four cases with *BRCA1* mutations; only one of those, involving a germline variant accompanied by somatic LOH, had a high HRD score. These results are consistent with data showing that in PC, biallelic alterations in *BRCA1* gene are less common than the genetic inactivation of the *BRCA2*, and accordingly, *BRCA1*-associated tumors are less likely to show signs of HRD [27].

PCs with *ATM* alterations were heterogeneous in terms of HRD-specific chromosomal instability. The median HRD score was higher in *ATM*-positive tumors (score = 24) than in cases without HRR alterations (score = 10) (*p* = 0.0005), but lower than in *BRCA2*-related cases (score = 41.5) (*p* = 0.005). Only three out of twelve (25%) PCs with putative biallelic *ATM* inactivation demonstrated HRD scores of at least 42; all these tumors had a combination of a germline pathogenic variant coupled with a second-hit somatic genetic event. These observations are in good agreement with the multiple data, which argue for low sensitivity of *ATM*-related PCs to PARPi [7,9].

Mutations in *CHEK2* and *NBN* accounted for 12% and 7% of all HRR-altered cases, respectively, and were predominantly represented by a few founder variants. PCs with *CHEK2* alterations were characterized by the lowest HRD scores (all below 20). Two out of eight (25%) *NBN*-related tumors had HRD; one of these cases had a combination of hereditary and somatic truncating variants.

High HRD scores were not detected in cases with *BLM*, *FANCA*, *FANCC*, *FANCI*, *RAD54L* or *MRE11* pathogenic variants. They were, however, observed in single patients with *FANCM* and *CDK12* inactivating alleles. Tumors with biallelic *CDK12* mutations constitute a distinct PC subtype. These tumors are known to be prevalent in metastatic castration-resistant prostate cancer, have a poor prognosis and exhibit a characteristic pattern of genome-wide tandem duplications [28,29]. In our cohort, 85% of cases with *CDK12* mutations exhibited the tandem duplication phenotype, including cases with single and double *CDK12* variants. Contrary to initial expectations, prostate cancer cases with *CDK12* mutations were found not to have typical HRD features; furthermore, there are several lines of evidences suggesting that *CDK12*-associated PCs demonstrate comparatively low sensitivity to PARP inhibitors [10,45,46,47]. It has been suggested that prostate cancer with *CDK12* mutations could be sensitive to immune checkpoint inhibitors due to the increased number of fusions; however, this hypothesis has not been confirmed by clinical data [48]. Nevertheless, recent animal study showed sensitivity of double *CDK12/TP53* knockout PC to immune checkpoint inhibitors [49]. Furthermore, CDK12 deficiency has been found to exhibit synthetic lethality when its paralog CDK13 is pharmacologically targeted, suggesting that this could be a promising novel treatment approach for this type of prostate cancer [49,50].

A substantial portion of PC cases without identified HRR mutations (5%) had a high HRD score. Large genomic studies have shown that up to one-third of all cases of HRD in metastatic PC are associated with gross biallelic deletions of the *BRCA2* gene, which are not detectable by conventional targeted NGS [14,51,52]. HRD may also be caused by epigenetic inactivation of HRR genes, functional mutations in non-coding regions, or alterations in genes not included in the 34 loci analyzed in this study [14]. Our results suggest that the analysis of chromosomal instability has a potential to expand the range of patients eligible for treatment with PARPi or platinum drugs.

## 4. Materials and Methods

The study group comprised 1131 prostate cancer patients referred to the N.N. Petrov National Medical Research Centre of Oncology between October 2022 and March 2025. Mutations in HRR genes were identified using the HRR35 targeted NGS panel (Nanodigmbio, Nanjing, China). CNV profiles were analyzed with the HiSNP Ultra Panel v1.0 (Nanodigmbio, Nanjing, China). The HRR35 panel includes all loci that are associated with administration of PARPi, as well as several other HRR genes (*BRCA1*, *BRCA2*, *ATM*, *PALB2*, *BARD1*, *RAD51*, *RAD51B*, *RAD51C*, *RAD51D*, *RAD54L*, *CDK12*, *BRIP1*, *CHEK2*, *FANCA, NBN*, *ATR*, *BLM*, *CHEK1*, *FANCC*, *FANCD2*, *FANCE*, *FANCF*, *FANCG*, *FANCI*, *FANCL*, *FANCM*, *MRE11*, *PPP2R2A*, *RAD50*, *RAD52*, *RBBP8*, *RPA1*, *SLX4, XRCC2*), and the *TP53* gene. The HiSNP panel enables the analysis of over 52,000 common single nucleotide polymorphisms (SNPs) with a resolution of around 50 Kb, making it suitable for determining the HRD score.

Participating physicians were encouraged to submit to the study both tumor and blood samples. Of the 1131 PCs, the material forwarded for targeted sequencing comprised paired blood and archival tumor samples in 962 cases, only archival histological material in 107 cases, and only blood samples in 62 cases (Figure 4). In instances where blood samples were not available, both tumor and normal tissues were manually microdissected from the archival histological material wherever possible. FFPE samples containing less than 10% tumor cells were excluded from the analysis of somatic HRR mutations and were not sequenced using the HiSNP Ultra Panel v1.0.

DNA extraction from FFPE samples was performed using the ExtractDNA FFPE reagent kit (Eurogen, Moscow, Russia). DNA library preparation for sequencing was carried out using the NadPrep EZ DNA Library Preparation Kit (Nanodigmbio, Nanjing, China) according to the manufacturer’s protocol. The concentration of DNA libraries was assessed using a Qubit fluorometer (Thermo Fisher Scientific, Waltham, MA, USA), and the fragmentation quality and average size were determined with a Fragment Analyzer 5200 (Agilent Technologies, Santa Clara, CA, USA). All 1131 studied cases were subjected to NGS using the HRR35 panel. For this purpose, DNA libraries with sufficient concentration (at least 500 ng) were pooled in sets of 12 for enrichment with HRR35 probes. In 680 cases where paired tumor and normal material was available, and where the concentrations of both libraries in each pair were sufficient for a second enrichment, hybridization was subsequently performed using the HiSNP Ultra Panel v1.0 probes. Enrichment with target probes was performed using the NadPrep Hybrid Capture Reagents kit (Nanodigmbio, Nanjing, China) according to the manufacturer’s protocol. After enrichment, the DNA libraries were circularized using the NadPrep Universal Circularization Kit (Nanodigmbio, Nanjing, China). DNA libraries were then sequenced in paired-end mode for 150 cycles in each direction using the DNBSEQ-50 instrument (MGItech, Shenzhen, China).

Primary bioinformatic processing of the sequencing data involved the following standard steps: generation of FASTQ files; quality control (QC); and alignment of the processed reads to the GRCh37 (hg19) human reference genome using the BWA aligner (version 0.7.18). The HaplotypeCaller software (GATK4) was then used to search for single nucleotide variants and small insertions/deletions. Somatic mutations in HRR genes were detected using the MuTect2 tool. DNA sequences were annotated using the SnpEff software tool (version 5.2c). Only somatic mutations with a depth of coverage greater than 10, detected in both the forward and reverse reads, were considered. Samples with a mean coverage of less than 100×, or with less than 80% of the target sequences covered at a depth of at least 30×, were excluded from the analysis. Detected HRR variants were classified as pathogenic or likely pathogenic if they had been already described in medical literature or ClinVar database, or if they represented previously undescribed or rare (<0.5% frequency in gnomAD population database) truncating germline variants. Somatic truncating variants were also considered pathogenic.

For germline variants, loss of heterozygosity (LOH) was assessed in the corresponding tumor, where available. LOH was confirmed if the tumor sample showed a variant allele frequency of more than 65% of the total reads.

Following sequencing with the HiSNP panel, somatic copy number variation (CNV) plots were constructed using the ACNV (Allelic Copy Number Variation) tool integrated into the GATK4 package [53]. The HRD score was calculated as the sum of the LOH, TAI and LST scores, defined as described in [54]. The threshold level for the presence of HRD was set at an HRD score of at least 42.

The Mann–Whitney test was used to compare the age of patients with and without HRR mutations, as well as those with high and low HRD scores. Associations for categorical data were assessed using Pearson’s chi-squared test, Fisher’s exact test or the Cochran–Armitage test for trend (for tumor size and stage). The odds ratios with the corresponding 95% confidence intervals were calculated to determine the strength of associations between the presence of HRR mutations or high HRD scores and clinical parameters. The Kruskal–Wallis test with Conover post hoc test was used to compare HRD score values in different categories of PC. A two-tailed p-value of less than 0.05 was considered statistically significant.

## 5. Conclusions

Approximately 10% of patients with aggressive PC carry germline pathogenic variants in genes with established roles in cancer predisposition *(BRCA2*, *ATM*, *CHEK2*, *NBN*, *BRCA1*, *PALB2*, *BRIP1*, *BARD1*). A significant proportion of PC with *BRCA2* mutations, including those with confirmed biallelic lesions, exhibit an HRD score below the commonly accepted threshold. Mutations in *BRCA2*, *PALB2* and *BRIP1* genes are associated with the highest level of chromosomal instability, whereas alterations in the other HRR loci (*CHEK2*, *NBN*, *BLM*, *FANCA*, *FANCC*, *FANCI*, *RAD54L*, *MRE11*, *CDK12*) are unlikely to result in HRD. PC with *ATM* mutations, including those with biallelic variants, exhibit high variability in HRD score values. The analysis of HRD score enables the identification of a significant number of PC cases, which are negative by HRR mutation testing but potentially sensitive to PARPi or platinum drugs.

Compared to targeted HRR genes analysis, HRD testing requires more powerful NGS equipment. Although it is routinely used in the treatment planning of ovarian cancer, HRD has not yet been incorporated into the management of breast and prostate carcinomas, probably because the availability of tumor tissue for these tumor entities is often limited to tiny biopsied material. Our study strongly suggests that attitudes towards selection of PC patients for PARPi treatment require significant revision. In fact, current versions of HRR tests often urge for the use of PARPi in those PC patients, who are unlikely to respond to this treatment, while they also miss a substantial portion of men, who show convincing features of potential tumor sensitivity to PARPi or some cytotoxic agents. Our data suggest that HRR gene panels require some modifications, e.g., the exclusion of CHEK2 and, perhaps, several other genes. At the same time, wherever possible, HRR tests should be supplemented by HRD assays, or, at least, by biallelic analysis of the genes affected by mutations.

## Figures and Tables

**Figure 1 ijms-26-11851-f001:**
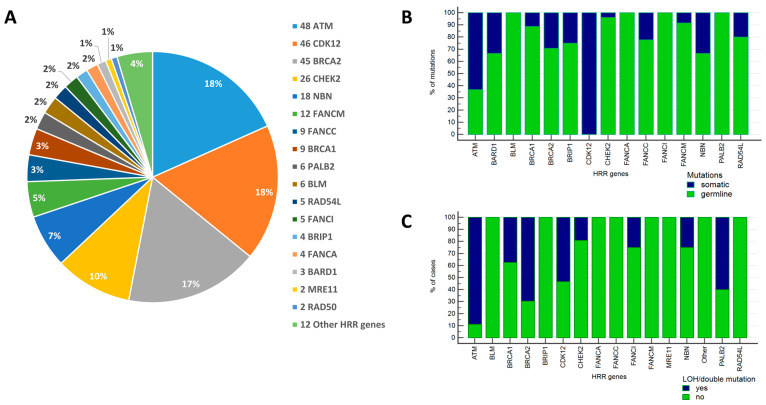
Spectrum of HRR mutations. (**A**) The share of mutations in various HRR genes out of a total of 262 identified mutations. The total number of PC cases with pathogenic variants for a given gene are indicated before the gene names. (**B**) The proportion of germline and somatic mutations detected in different HRR genes. For the majority of genes (except *CDK12* and *ATM*), germline mutations prevailed over somatic alterations. (**C**) The proportion of PC cases with LOH or double mutations (defined as a combination of a germline and somatic mutation, or two somatic mutations in the same gene) for different HRR genes. The highest shares of LOH/double mutations were observed in cases with *ATM*, *BRCA2*, *PALB2*, *CDK12* and *BRCA1* mutations.

**Figure 2 ijms-26-11851-f002:**
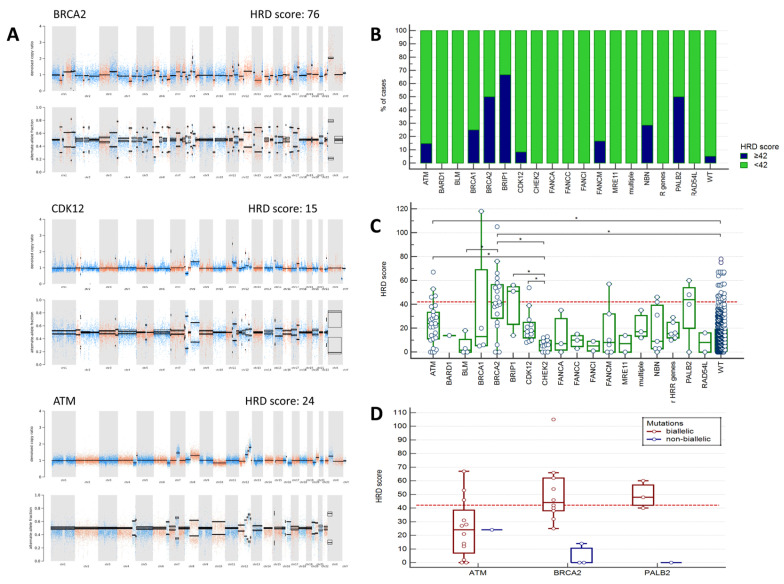
HRD score in PC cases with various HRR mutations. (**A**) Examples of CNV profiles in PCs with *BRCA2, CDK12* and *ATM* mutations. In each case, the upper part of the image shows the normalized sequencing read counts for each analyzed heterozygous SNP across all chromosomes. The bottom part shows the alternative allele fractions for the same SNPs. (**B**) The proportion of tumors with an HRD score ≥ 42 among cases with mutations in different HRR genes and in PC without HRR mutations (WT). PCs with *BRCA2*, *BRIP1* and *PALB2* mutations had a high HRD score in at least 50% of the analyzed cases. (**C**) Distribution of HRD score values in PCs with mutations in different HRR genes and in cases without HRR alterations (WT). The blue dots represent individual cases. The lower and upper borders of the green boxes correspond to the first and third quartiles, respectively, while the whiskers show the minimum and maximum HRD score values. Statistically significant differences between groups with various mutations are marked with asterisks (*). (**D**) HRD score values in tumors with biallelic or monoallelic inactivation of the *ATM*, *BRCA2* or *PALB2* genes. In panels (**C**,**D**), the dashed red horizontal line corresponds to an HRD score of 42.

**Figure 3 ijms-26-11851-f003:**
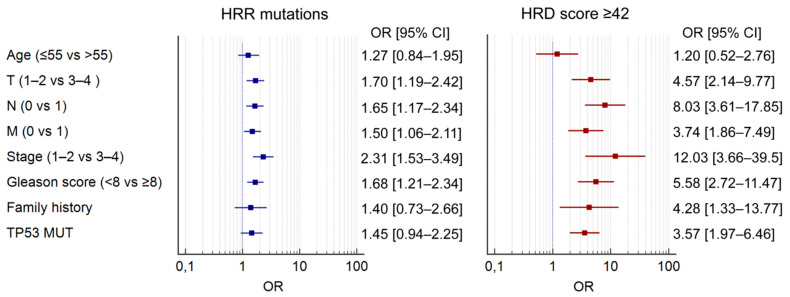
Forest plots showing the odds ratios (OR) and 95% confidence intervals (CI) for the occurrence of HRR mutations (left) and a high HRD score (right), depending on various clinical parameters. Evidently higher OR values were observed for tumors with the HRD score ≥ 42 as compared to PCs with HRR mutations across all analyzed parameters except the age.

**Figure 4 ijms-26-11851-f004:**
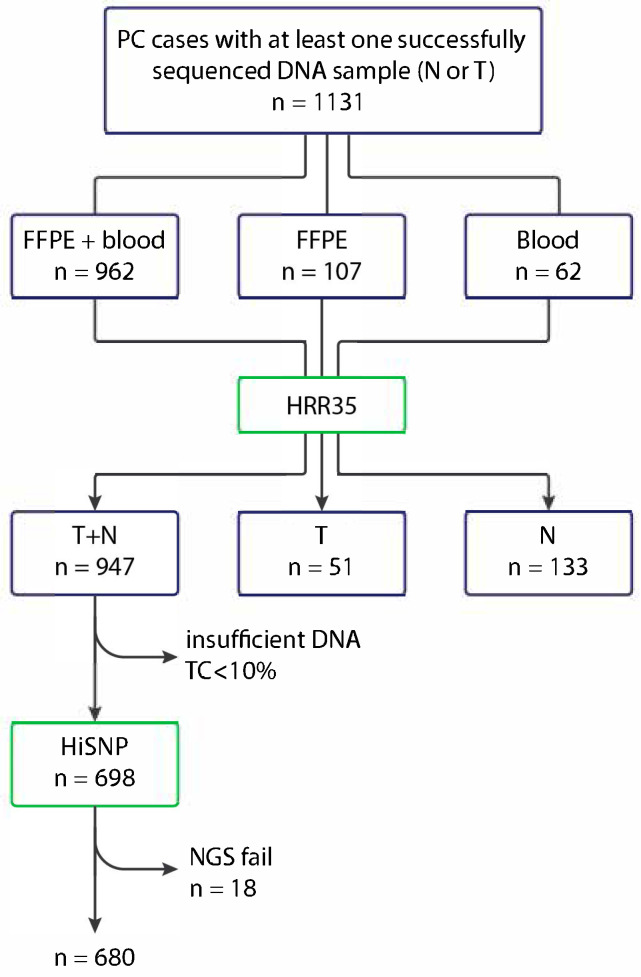
Flow-chart representing the scheme of the study. The initial collection of 1131 PC cases was represented by paired tumor and archival histological material, archival tissue only, or blood samples only. Next-generation sequencing (NGS) using the HRR35 panel generated data for paired normal and tumor material in 947 cases, tumor tissue only in 51 cases, and normal DNA only in 133 cases. Of the 947 available paired tumor-normal DNA samples, 698 were suitable for NGS using the HiSNP panel, with success achieved in 680 cases. FFPE—formalin-fixed paraffin-embedded archival tissues; N—normal; T—tumor; TC—tumor cell content of histological material.

**Table 1 ijms-26-11851-t001:** Clinicopathological characteristics of PCs tested for HRR mutations and HRD score.

Characteristic	PC Cases Tested for HRR Mutations (*n* = 1131)	PC Cases Tested for HRD Score (*n* = 680)
Mean age at diagnosis; years (age range)	64.6 (40–87)	64.9 (41–85)
Cases with age at diagnosis ≤ 55 years	146 (12.9%)	79 (11.6%)
Tumor size (T)		
T1	169 (14.9%)	157 (23.1%)
T2	205 (18.1%)	105 (15.4%)
T3	305 (27.0%)	141 (20.7%)
T4	182 (16.1%)	95 (14.0%)
Nd *	270 (23.9%)	182 (26.8%)
Lymph node status (N)		
N0	468 (41.4%)	305 (44.9%)
N1	362 (32.0%)	180 (26.5%)
Nd	301 (26.6%)	195 (28.7%)
Distant metastases (M)		
M0	438 (38.7%)	275 (40.4%)
M1	403 (35.6%)	206 (30.3%)
Nd	290 (25.6%)	199 (29.3%)
Stage		
1	165 (14.6%)	152 (22.4%)
2	117 (10.3%)	61 (9.0%)
3	75 (6.6%)	28 (4.1%)
4	477 (42.2%)	238 (35.0%)
Nd	297 (26.3%)	201 (29.6%)
Gleason score		
<8	566 (50.0%)	397 (58.4%)
≥8	397 (35.1%)	204 (30.0%)
Nd	168 (14.9%)	79 (11.6%)

* Nd: no data.

**Table 2 ijms-26-11851-t002:** Associations between the presence of HRR mutations or high HRD score and clinicopathological characteristics of PCs.

	HRR Mutations	Significance, *p*-Value	HRD Score ≥ 42	Significance, *p*-Value
Age at diagnosis				
≤55	33/146 (22.6%)	0.251	7/79 (8.9%)	0.666
>55	183/984 (18.6%)		45/601 (7.5%)	
Tumor size (T)				
T1	21/169 (12.4%)	0.022	1/157 (0.6%)	<0.0001
T2	35/205 (17.1%)		8/105 (7.6%)	
T3	78/304 (25.7%)		16/141 (11.3%)	
T4	34/182 (18.7%)		17/95 (17.9%)	
Nodal involvement (N)				
N0	74/467 (15.8%)	0.004	8/305 (2.6%)	<0.0001
N1	86/362 (23.8%)		32/180 (17.8%)	
Distant metastases (M)				
M0	72/437 (16.5%)	0.020	12/275 (4.4%)	0.0001
M1	92/403 (22.8%)		30/206 (14.6%)	
Tumor stage				
1	20/165 (12.1%)	0.0002	1/152 (0.7%)	<0.0001
2	13/117 (11.1%)		2/61 (3.3%)	
3	18/74 (24.3%)		1/28 (3.6%)	
4	111/477 (23.3%)		38/238 (16.0%)	
Family history of cancer				
Negative or no data	203/1077 (18.8%)	0.305	48/664 (7.2%)	0.028
Positive	13/53 (24.5%)		4/16 (25.0%)	
Gleason grade				
<8	84/566 (14.8%)	0.002	11/397 (2.8%)	<0.0001
≥8	90/397 (22.7%)		28/204 (13.7%)	
TP53 somatic mutation				
WT	163/823 (19.8%)	0.090	31/558 (5.6%)	<0.0001
MUT	28/193 (14.5%)		21/121 (17.4%)	

## Data Availability

The original contributions presented in this study are included in the article/Appendix A. Further inquiries can be directed to the corresponding authors.

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
