# Peer review of "Complex Relationships Between Homologous Recombination Deficiency (HRD) Score and Mutational Status of Homologous Recombination Repair (HRR) Genes in Prostate Carcinomas"

_ijms, 2025, doi:10.3390/ijms262411851_

Round 1
Reviewer 1 Report
Comments and Suggestions for Authors
The article "Complex relationships between homologous recombination deficiency (HRD) score and mutational status of homologous recombination repair (HRR) genes in prostate carcinomas" by A. G. Iyevlev et al. is a successful attempt to analyze the relationship between homologous recombination deficiency-causing mutations, potentially leading to genetic instability, which predisposes to PARP inhibitors therapy in prostate cancers. The experiment was well-designed, conducted, and described. The authors used adequate statistical methods for data analysis. It will be of great interest to readers researching predictive factors in the treatment of aggressive prostate cancers.
Author Response
Thank you for your positive evaluation of our work.
Reviewer 2 Report
Comments and Suggestions for Authors
This is a relevant article in its field of study, as it focuses on a specific population. Classifying patients by HDR score can benefit patients and reduce treatment costs by enabling them to receive an appropriate treatment.
However, it is sometimes difficult to follow, with very long sentences and a confusing presentation of patient numbers. Initially, they stated that the study included 1,131 patients. but the HRD score could only be calculated for 680. It is unclear what type of analysis was performed to calculate the HRD score for these 680 patients. In one part, they report detecting mutations in HRR genes in 216 patients; in another, they report conducting genetic studies in 120 patients. Please clarify.
Another point is that they do not mention why they set their cutoff point at an HRD score of 42; please explain.
It seems to me that the supplementary oncoplot figure should be included in the body of the article.
There is a need to discuss your findings in comparison with other populations.
It is recommended to use the standard nomenclature for human genes (https://www.genenames.org/) in text and figures.
A grammar review is recommended.
Author Response
Comment 1. However, it is sometimes difficult to follow, with very long sentences and a confusing presentation of patient numbers. Initially, they stated that the study included 1,131 patients. but the HRD score could only be calculated for 680. It is unclear what type of analysis was performed to calculate the HRD score for these 680 patients. In one part, they report detecting mutations in HRR genes in 216 patients; in another, they report conducting genetic studies in 120 patients. Please clarify.
Response:
The numbers of cases examined for the presence of HRR mutations and for the HRD scores, as well as the corresponding methods, are now clarified at the beginning of the two Results subsections:
2.1. Frequency and spectrum of mutations in HRR genes
The coding sequences of the 34 HRR genes were analyzed in 1131 prostate cancer cases using targeted NGS analysis.
…
2.2. Analysis of HRD scores
Out of the total 1131 studied cases, HRD testing was successfully completed for 680 patients, for whom the paired tumor and normal DNA of sufficient quality and quantity were available. In these cases, HRD scores were calculated as a sum of LOH, LST and TAI measures obtained after targeted NGS analysis with the HiSNP Ultra Panel v1.0 (Nanodigmbio, China). An HRD score of 42, which is commonly accepted as a clinically valid threshold for ovarian and breast cancer, was used to determine the presence of HRD [19-23].
The legend for Figure 4 with the flow-chart of the study was extended:
Figure 4. Flow-chart representing the scheme of the study. The initial collection of 1131 PC cases was represented by paired tumor and archival histological material, archival tissue only, or blood samples only. Next-generation sequencing (NGS) using the HRR35 panel generated data for paired normal and tumor material in 947 cases, tumor tissue only in 51 cases, and normal DNA only in 133 cases. Of the 947 available paired tumor-normal DNA samples, 698 were suitable for NGS using the HiSNP panel, with success achieved in 680 cases.
The detailed description of the NGS sequencing panels and protocols, and the criteria for sample exclusion from the study, are presented in the Methods section and in Figure 4. The two distinct figures of HRR-mutated cases (216 and 120) correspond to the first and second parts of the study, i.e. to the whole PC sample (n = 1131) and to the PC subgroup analyzed for HRD score (n = 680). We hope that adding explanatory phrases to the results has made this more obvious.
Comment 2. Another point is that they do not mention why they set their cutoff point at an HRD score of 42; please explain.
Response:
Thank you for this valuable comment. The rationale for choosing a threshold value of 42 is now provided in the Results section:
An HRD score of 42, which is commonly accepted as a clinically valid threshold for ovarian and breast cancer, was used to determine the presence of HRD [19-23].
Comment 3. It seems to me that the supplementary oncoplot figure should be included in the body of the article.
Response:
We agree that Figure S1, given its comprehensive nature, is well-suited for inclusion in the main text of the article. Nevertheless, we have opted to place it in the Supplementary materials to enhance its readability and visual accessibility, which may be challenging to accomplish in the printed format due to its considerable size.
Comment 4. There is a need to discuss your findings in comparison with other populations.
Response:
Discussion section was extended and now contains the following paragraph:
Analysis of our data in comparison to other studies suggests that there may be interethnic variations in the distribution of HRR mutations in PC. We observed lower incidences of BRCA2 and CDK12 mutations (42/1131; 3.7% and 30/1131; 2.7%, respectively) than those reported in large European and American studies, which produced the rates of 9–12% for BRCA2 and around 7% for CDK12 [31, 37, 38]. Conversely, our dataset revealed a significant prevalence of CHEK2 and NBN alterations, accounting for 2.3% and 1.3% of analyzed PCs, respectively. The latter is likely to be a consequence of the high frequency of several founder CHEK2 and NBN cancer-predisposing alleles in Slavic populations. Within our study group, 66% of individuals with HRR mutations had germline alterations. In contrast, Olmos et al. (2025) reported a considerably lower percentage of germline variants (32%). Along with the distinct ethnic composition of the studied cohorts, these differences could be attributed to the varying clinical characteristics of the patients, non-identical HRR gene panels, testing of either tumor alone or paired tumor and normal tissues, and differences in the interpretation of variant pathogenicity. The reliability of somatic mutation detection also depends heavily on the proportion of tumor cells in the sample and the sequencing depth. We employed a fairly lenient threshold of 10% tumor cells in a tissue sample, which could potentially impact the identification of somatic variants.
Comment 5. It is recommended to use the standard nomenclature for human genes (https://www.genenames.org/) in text and figures.
Response:
We have checked the gene names used in the manuscript but did not find any inconsistencies with the standard nomenclature.
Comment 6. A grammar review is recommended.
Response:
The grammar review was performed.
Reviewer 3 Report
Comments and Suggestions for Authors
This manuscript presents a valuable and comprehensive study analyzing homologous recombination repair (HRR) gene mutations and homologous recombination deficiency (HRD) scores in over a thousand prostate cancer cases. The large sample size and inclusion of paired tumor-normal sequencing provide strong statistical validity. The investigation of 34 HRR genes extends beyond the usual altered genes such as BRCA1/2, offering novel insights into the mutation spectrum.
The study’s integration of genomic mutation data with chromosomal instability measures (HRD scores) is highly relevant clinically, especially for predicting responsiveness to PARP inhibitors (PARPi). The statistical analyses are rigorous, and the clinical implications addressed—particularly the recommendation to revise existing HRR gene panels—are timely and important.
Several aspects could be improved to enhance clarity and flow.
First, the discussion around the HRD score cut-off would benefit from elaboration; the authors suggest the conventional threshold of 42 used in ovarian and breast cancer may not be suitable in prostate cancer but do not propose alternatives or thresholds based on their data.
Second, more detailed acknowledgement of tumor heterogeneity, sample quality, and sequencing depth effects on mutation and HRD detection would add nuance, especially since some samples lacked paired normal or tumor DNA. Additionally, the manuscript would gain strength by discussing the clinical treatment background of patients, as prior therapy could influence genomic profiles, particularly in metastatic cancers. The figures and tables are informative but would be easier to interpret with more descriptive legends emphasizing clinical significance and data subsets.
Lastly, minor typographical and formatting refinements would improve professionalism.
The findings about specific gene alterations are notable.
The low HRD scores associated with CHEK2 mutations support existing literature and indicate such genes might be excluded from PARPi panels. CDK12 mutations are reliably linked to tandem duplication phenotypes, confirming prior reports. The differentiation between monoallelic and biallelic mutations in genes like BRCA2 and PALB2 correlating with HRD score nuances is clinically meaningful. The detection of a previously unrecognized CDK12 deletion underscores the value of comprehensive genomic analyses.
Overall, this manuscript substantially advances understanding of HRR mutations and HRD in prostate cancer, with direct implications for improving PARPi therapy selection.
Author Response
Comment 1. First, the discussion around the HRD score cut-off would benefit from elaboration; the authors suggest the conventional threshold of 42 used in ovarian and breast cancer may not be suitable in prostate cancer but do not propose alternatives or thresholds based on their data.
Response:
The discussion around the HRD score cut-off was extended:
… This threshold is clinically significant in ovarian and breast cancer, allowing the detection of 95% of BRCA1/2-positive tumors of these types and identifying patients who may benefit from PARP inhibitors and platinum drugs [19-23]. Our data, along with the results of several similar studies, suggest that HRD scores tend to be lower in PCs, including BRCA2-associated tumors [24, 26, 27]. One of the possible explanations is the lower occurrence of somatic TP53 mutations in PC. Indeed, TP53 alterations are known to be strongly associated with elevated chromosomal instability [24, 26, 39], and our study confirmed a strong correlation between TP53 gene status and high HRD (Table 2; Figure 3). We also examined whether the low HRD scores obtained in BRCA2- and BRCA1-mutated cases in our study are attributed to suboptimal tumor cell content in the corresponding tissue samples. However, we concluded that this factor was unlikely to have influenced the results in our series, since the BRCA2-mutated PC with the lowest observed tumor cell content (15%) demonstrated a high HRD score of 76 (data not shown). Overall, in our dataset 95% of cases with confirmed biallelic inactivation of BRCA2, BRCA1 or PALB2 had an HRD score of at least 25, whereas only 60% of such tumors demonstrated a score ≥ 42. Taken together, these data indicate that the optimal threshold for HRD score in PC may indeed differ from the one accepted for ovarian/breast carcinomas. Future studies have to address a predictive significance of various HRD cut-offs for tumor sensitivity to PARPi and other drugs.
Comment 2. Second, more detailed acknowledgement of tumor heterogeneity, sample quality, and sequencing depth effects on mutation and HRD detection would add nuance, especially since some samples lacked paired normal or tumor DNA.
Response:
The paragraphs discussing the mentioned issues were added:
Along with the distinct ethnic composition of the studied cohorts, these differences could be attributed to the varying clinical characteristics of the patients, non-identical HRR gene panels, testing of either tumor alone or paired tumor and normal tissues, and differences in the interpretation of variant pathogenicity. The reliability of somatic mutation detection also depends heavily on the proportion of tumor cells in the sample and the sequencing depth. We employed a fairly lenient threshold of 10% tumor cells in a tissue sample, which could potentially impact the identification of somatic variants.
...
We also examined whether the low HRD scores obtained in BRCA2- and BRCA1-mutated cases in our study are attributed to suboptimal tumor cell content in the corresponding tissue samples. However, we concluded that this factor was unlikely to have influenced the results in our series, since the BRCA2-mutated PC with the lowest observed tumor cell content (15%) demonstrated a high HRD score of 76 (data not shown).
Comment 3. Additionally, the manuscript would gain strength by discussing the clinical treatment background of patients, as prior therapy could influence genomic profiles, particularly in metastatic cancers.
Response:
We acknowledge that the data on clinical background of the patients would be valuable for interpretation of the study results, however, unfortunately, we have no access to this information.
Comment 4. The figures and tables are informative but would be easier to interpret with more descriptive legends emphasizing clinical significance and data subsets.
Response:
All figure legends were revised and extended and now emphasize the main findings.
Comment 5. Lastly, minor typographical and formatting refinements would improve professionalism.
Response:
The manuscript was checked for typographical or grammatical errors.